# Human Body Mixed Motion Pattern Recognition Method Based on Multi-Source Feature Parameter Fusion

**DOI:** 10.3390/s20020537

**Published:** 2020-01-18

**Authors:** Jiyuan Song, Aibin Zhu, Yao Tu, Yingxu Wang, Muhammad Affan Arif, Huang Shen, Zhitao Shen, Xiaodong Zhang, Guangzhong Cao

**Affiliations:** 1Institute of Robotics & Intelligent Systems, Xi’an Jiaotong University, Xi’an 710049, China; jysong@stu.xjtu.edu.cn (J.S.); tu1007909971@stu.xjtu.edu.cn (Y.T.); wyx13596009525@stu.xjtu.edu.cn (Y.W.); affanarif025@hotmail.com (M.A.A.); shenhuang@stu.xjtu.edu.cn (H.S.); shen940123@stu.xjtu.edu.cn (Z.S.); xdzhang@mail.xjtu.edu.cn (X.Z.); 2Shaanxi Key Laboratory of Intelligent Robots, Xi’an 710049, China; 3Key Laboratory of Education Ministry for Modern Design and Rotor-Bearing System, Xi’an 710049, China; 4Shenzhen Key Laboratory of Electromagnetic Control, Shenzhen University, Shenzhen 518060, China; gzcao@szu.edu.cn

**Keywords:** motion pattern recognition, plantar pressure, inertial sensor, neural network, lower limb assisted exoskeleton

## Abstract

Aiming at the requirement of rapid recognition of the wearer’s gait stage in the process of intelligent hybrid control of an exoskeleton, this paper studies the human body mixed motion pattern recognition technology based on multi-source feature parameters. We obtain information on human lower extremity acceleration and plantar analyze the relationship between these parameters and gait cycle studying the motion state recognition method based on feature evaluation and neural network. Based on the actual requirements of exoskeleton per use, 15 common gait patterns were determined. Using this, the studies were carried out on the time domain, frequency domain, and energy feature extraction of multi-source lower extremity motion information. The distance-based feature screening method was used to extract the optimal features. Finally, based on the multi-layer BP (back propagation) neural network, a nonlinear mapping model between feature quantity and motion state was established. The experimental results showed that the recognition accuracy in single motion mode can reach up to 98.28%, while the recognition accuracy of the two groups of experiments in mixed motion mode was found to be 92.7% and 97.4%, respectively. The feasibility and effectiveness of the model were verified.

## 1. Introduction

In order to achieve compliant human computer interaction and coordinated motion, the exoskeleton control strategy develops from a passive position control mode to a mixture of multiple control modes, and the correct recognition of the gait is the basis of the hybrid control of the exoskeleton robot. As a key technical link in the flexible motion control of exoskeleton robots, gait analysis is a technique based on the acquisition, description, and analysis of human kinematics, dynamics and physiological information to understand the laws of human motion and walking mechanism [1]. In indoor environments, motion state recognition based on the pictures and videos [2,3], and the force platform are the most accurate systems for gait analysis. Wearable gait systems are comparably small, durable, flexible and adaptable, capable of being used in complex outdoor environments [4]. The data acquisition method of the wearable gait analysis system is simple and environment-free, and is easy to integrate with the exoskeleton. During the interactive control, the dynamic measurement results of the human lower limb motion posture data are provided in real time, and the exoskeleton is assisted to obtain more gait learning samples to decode, quickly and accurately, the collected lower extremity motion information for obtaining the current motion state of the wearer. The exoskeleton can be trained to switch the movement mode autonomously according to the recognized posture or movement pattern of the human body, making its movement more flexible to adapt to more application scenarios, and the obtained human motion information can also be used to rectify the results of other perception methods of motion intentions (such as EEG and EMG). In addition, gait phase division is also important in the diagnosis and prevention of pathological gait [5], medical monitoring [6,7], and of rehabilitation effect evaluation [8]. The key to the feat of motion state identification is the feature extraction and the expression of the lower limb motion information, and the realization of multi-pattern classification algorithm [9]. A good recognition model can obtain higher recognition with lower computational load in a shorter time with better precision.

A.M. Khan et al. [10] used accelerometers to collect data, using the autoregressive coefficients of motion signals as key features, and achieved 99% accuracy for the four movements of standing and lying. Baojun Chen et al. [11] studied discrete the application of the plantar pressure distribution signal in motion pattern recognition; Andrea Parri et al. [12] proposed a hybrid classification method mainly used for real-time motion pattern recognition of lower extremity wearable exoskeleton robots. Trung Thanh Ngo et al. [13] performed a pattern recognition experiment of continuous motion signals obtained from the waist of the subjects using tri-axial accelerometer. Liu Lei et al. [14] used the foot pressure information to decompose the lower limbs of the human body and identified three kinds of movements, namely: flat walking, upstairs and downstairs, by employing the generalized regression neural network (GRNN). Tkach et al. [15] and Hargrove et al. [16] combined surface electromyography (sEMG) and mechanical sensor, used pattern recognition algorithm to decode EMG signal, and combined with data from prosthesis sensor, realized seamless transition between walking on flat ground, stairs, and ramps.

We chose gyroscope and force sensor to collect human motion data because they are smaller and easier to integrate with exoskeleton compared with image acquisition equipment and EMG acquisition equipment. In general, data related to acceleration, angular velocity, and plantar pressure are most commonly used for gait phase segmentation. At present, the related research mainly focuses on a single movement or posture scene of the human body. In order to improve the flexibility of exoskeleton movement, increasing the number of scene recognition categories and the correct rate is still the key to motion state recognition. In this paper, we first determined the 15 most common gait patterns; such as standing still, standing with weight, sitting, one knee down, fast walking, constant speed walking, slow walking, walking in place, running, stepping up continuously, stepping down continuously, single step stepping up, single step stepping down, uphill and downhill; based on the actual application requirements of an exoskeleton. Then, a study of the feature extraction and feature screening methods of multi-source motion information fusion was carried out, and analyzed various motion behavior characteristics from various original motion data. A high-dimensional feature vector matrix was constructed by extracting motion features that can describe and distinguish various motions. Then, based on the distance-based feature selection method, the high-dimensional feature vector matrix obtained from the acquired high-dimensional feature vector matrix is selected. Based on the multi-layer BP neural network, a nonlinear mapping model between feature quantity and motion state was established. Finally, the feasibility and effectiveness of the model to accurately identify the motion state were verified by experiments.

## 2. Materials and Methods

### 2.1. Experimental Principles and Systems

The hip, knee, and ankle joints are the pivots of the lower extremity limbs, mainly in the sagittal plane, and it is important to quantify these joint angles. The foot is the end of the limb, and the foot movement is the result of the joint movement of the lower limbs. Therefore, the foot pressure information, which is one of the main components of the walking parameters, has important research value. In order to fully describe the movement of the lower limbs, a gait analysis system is proposed to combine the inertial sensor and the pressure sensor to perform the synchronous acquisition of the motion information of the thigh, the calf and the foot and the pressure information of the sole.
(1)$$\{\begin{array}{c} \theta_{hip}=\theta_{thigh}\\ \theta_{knee}=\theta_{shank}-\theta_{thigh}\\ \theta_{ankle}=\theta_{foot}-\theta_{shank} \end{array}$$

Inertial measurement unit (IMU), including accelerometers, gyroscopes, and magnetometers, is the most widely used wearable sensors in clinical research and are often used in place of professional optical motion capture systems [17,18,19,20]. The principle of measuring the hip and knee angle corresponding to the inertial sensor arrangement is shown in Figure 1 and Formula (1).

The wearable insole, with built-in pressure sensors, is designed to easily measure the interaction between the sole and the ground and is used to detect foot events and body weight distribution [21,22,23,24]. The pressure distribution of the foot region measured by the Novel Pedar-X insole foot pressure distribution measurement system is shown in Figure 2a. Taking the 65 kg subject as an example, the pressure curve of each region during a gait cycle is plotted, as shown in Figure 2b. The main focus is on the thumb area (T area), the first metatarsal area (M1 & M2 areas), and the heel area (HM, HL & HC areas), so we arrange the pressure sensors for these three areas. As shown in Figure 3, the positions of the force sensors, we chose, for the left foot are a, b, and c, while that for the right foot are e, f, and g.

The overall design of the wearable gait analysis system is shown in Figure 3. It includes a total of 3 IMUs at the thigh, calf and foot of each leg, and 3 force sensors at the sole of each foot. The foot used in our system is flexible as, it is part of an exoskeleton [25]. We transformed and corrected the data collected by the sensor, which is very accurate compared with the data collected by Vicon motion capture system. This is explained in detail in the results section, which proves the effectiveness and validity of our research tools.

### 2.2. Selection of Common Gait Patterns

According to our common use of exoskeleton, in this paper we refer to the assisted exoskeleton challenge program, we selected 15 common human lower limb movement modes, including 4 static modes and 11 dynamic modes, covering three environments of the road surface i.e., flat roads, stair surfaces, and slopes. The specific action points of various motion states are described in detail in Table 1.

### 2.3. Feature Parameter Extraction

#### 2.3.1. Time Domain Features and Frequency Domain Characteristics

The walking motion of the human body is usually a quasi-periodic process. Therefore, by calculating the frequency domain characteristics of the signal to obtain the frequency components in the motion signal and the energy level of each frequency component, it is helpful to realize the accurate recognition of the human motion behavior. Feature extraction is an extremely important part of pattern recognition. The quality of feature extraction will have direct consequence on the accuracy and computational complexity of recognition. This paper has integrated time domain feature parameters, frequency domain feature parameters and energy feature parameters with motion data; performing an accurate and comprehensive description to reflect the changes in the state of motion of the human body. The mean, variance, minimum value, correlation coefficient, and Fourier series are selected from the time domain and the frequency domain features.

#### 2.3.2. Energy Domain Characteristics

In the energy characteristic parameters, we selected the magnitude of the acceleration signal and the wavelet energy entropy. The integral of the triaxial acceleration vector with respect to time, SMA (the magnitude of the acceleration signal), in a certain period of time can indirectly reflect the energy consumption of the human body during the period of time, which is very different in various types of activities of the human body and act as a parameter to distinguish between human activity and static state. Wavelet analysis is a commonly used time-frequency analysis tool in the field of pattern recognition. The feature signal obtained after wavelet decomposition contains more time domain information in addition to frequency domain information. The wavelet entropy can detect the motion mutation point well and can be used as a feature for classification.

### 2.4. Distance-Based Feature Evaluation Method and Selection

For the eigenvalues extracted based on experience, the extracted feature quantities are not sensitive to the motion patterns. In order to simplify the model structure, improve the computing speed and generalization ability, we need to further filter the feature values for the classification. The result is a sufficiently necessary subset of features.

For the high-sensitivity feature, the distance between sample points of different categories in the feature space is as large as possible, and the distance between sample points within the same category is as small as possible [26]. Using this as a criterion, the ratio of the average distance between the classes in the training sample set and the average distance within the class can be calculated. The calculation process is shown in Figure 4. The large evaluation factor means a more dense intra-group distribution and a more discrete group distribution, and the trained classifier makes it easier to distinguish between multiple categories.

Suppose that the number of categories is *C* and the number of features is *J*’s feature set *q*.
(2)$$\left\{q_{m,c,j},m=1,2,\ldots M_{c};c=1,2\ldots ,C;j=1,2,\ldots J\right\}$$
where: $q_{m,c,j}$ refers to —The *j*th feature of the mth sample of class *C*; $M_{c}$—The total number of samples of class *C*; J—The number of samples for each class.

Calculating the intra-class average distance of all samples in each category, and then determining the average of the distances within *C* classes:(3)$$d_{c,j}=\frac{1}{M_{c}\times (M_{c}-1)}\sum_{m=1}^{M_{c}}\sum_{l=1}^{M_{c}}\left|q_{m,c,j}-q_{l,c,j}\right|,l,m=1,2,\cdots ,M_{c},l\neq m$$
(4)$$d_{j}^{\left(w\right)}=\frac{1}{C}\sum_{c=1}^{C}d_{c,j}$$

The intra-class distance difference factor can be calculated by using (5) for each dimension feature based on the maximum and minimum values of the distance within the class.
(5)$$v_{j}^{\left(w\right)}=\frac{\max(d_{c,j})}{\min(d_{c,j})}$$

Calculating the average of each feature of all samples of the same class is equivalent to determining the position of the center point of each category in the dimension of the *j* feature and determining the average distance between the different classes of the *j* feature dimensions.
(6)$$u_{c,j}=\frac{1}{M_{c}}\sum_{m=1}^{M_{c}}q_{m,c,j}$$
(7)$$d_{j}^{\left(b\right)}=\frac{1}{C\times (C-1)}\sum_{e=1}^{C}\sum_{c,e=1}^{C}\left|u_{c,j}-u_{e,j}\right|,c,e=1,2,\cdots ,C,c\neq e$$

Defining and calculating the difference factor between distances between classes as:(8)$$v_{j}^{\left(b\right)}=\frac{\max\left(\left|u_{e,j}-u_{c,j}\right|\right)}{\min\left(\left|u_{e,j}-u_{c,j}\right|\right)},c,e=1,2,\cdots ,C,c\neq e$$

In order to improve the ability of feature evaluation, the weighting factor was determined by the difference factor between the intra-class distance and the difference factor between the classes, and the inter-class and intra-class distance ratios with weighting factors are calculated.
(9)$$\lambda_{j}=\frac{1}{\frac{v_{j}^{\left(w\right)}}{\max(v_{j}^{\left(w\right)})}+\frac{v_{j}^{\left(b\right)}}{\max(v_{j}^{\left(b\right)})}}$$
(10)$$\alpha_{j}=\lambda_{j}\frac{d_{j}^{\left(b\right)}}{d_{j}^{\left(w\right)}}$$

Finally, the maximum normalization of $\alpha_{j}$ was obtained to obtain the normalized distance estimation factor.
(11)$${\overset{-}{\alpha }}_{j}=\frac{\alpha_{j}}{\max(\alpha_{j})}$$

### 2.5. Motion State Recognition Model

The working process of neural network can mainly be divided into two parts: training and testing. The motion pattern recognition model was established by combining the feature evaluation method and the BP neural network. The training process is shown in Figure 5, and the test flow is shown in Figure 6. In this paper, the hierarchical recognition model is adopted. On the one hand, the calculation amount is reduced by reducing the number of eigenvalues required for single-point discrimination. On the other hand, by reducing the input and output of the model, the recognition model is simplified, the training time of the model is shortened, and the recognition is improved. Robustness lays the foundation for online motion pattern recognition.

## 3. Results

### 3.1. Commissioning and Calibration of the Experimental System

The test subjects wore this system to walk, we did signal calibration and noise reduction filtering processing for the sensors, and compared the data from the hardware and the Vicon Vero2.2 infrared 3D motion capture system for time normalization. As shown in Figure 7, the data of the hip and knee angles obtained by the two systems during a uniform walking period for 25 s shows the overall trend of the joint angle information acquired by the wearable system and the Vicon system in the 25 s acquisition experiment. The average hip error was 1.254° ± 0.270°, the maximum error was 4.145° ± 0.856°, the correlation coefficient was 0.996 ± 0.002, the average knee error was 3.296° ± 1.295°, and the maximum error was 11.94° ± 1.826°. The cross-correlation coefficient is 0.973 ± 0.006, and there was no obvious cumulative error during long-time acquisition, which can be used for subsequent gait data acquisition and gait parameter calculation.

### 3.2. Procedure

We performed the four movement modes of standing, descending stairs, constant speed walking and fast walking in a sequence and collected gait data. As shown in Figure 8, the hip joint angle and the *three-axis* acceleration of the foot during the period were changed. The preprocessing and windowing interception operations were performed to obtain the variation law of various feature values with time as the data window slides, and the sensitivity of various features to the motion state is analyzed accordingly. The result is shown in Figure 9.

It can be seen that various features have their own advantages and disadvantages in the description of the motion mode. Therefore, the 141-dimensional features will be extracted according to the acquired multi-source information. The specific feature names and numbers are shown in Table 2. In order to ensure the real-time and safety of the human body in the actual use of the exoskeleton, the action sequence should be recognized as continuously as possible for the acquired action sequence and ensure high accuracy. Before the feature extraction, the collected data needs to be preprocessed and windowed separately. The sampling frequency is 60 Hz and the window length is set to $T_{w}=64$ samples. The time window sliding process needs to overlap each other to ensure the continuity of the calssification. It is generally considered that the human action frequency is low, and generally remains unchanged within one second, so the stacking length is selected as $T_{s}=32$ samples.

According to experience, there must be a certain number of weak correlations or redundant features in the 141-dimensional features. Before the classification model training, feature selection of high-dimensional feature sets is needed.

The distance evaluation factors of each feature quantity were obtained by distance evaluation between the dynamic feature set and the static feature set, and the calculation result is shown in Figure 10. It can be seen that in the static posture of the human body, the larger evaluation factors are the average and the maximum value, and the characteristics of the fluctuations of other reaction data such as the variance and the acceleration amplitude are small, which is consistent with the actual experience. Different static postures have different leg angles, and the accelerations perceived by the sensitive axes of the acceleration sensor are greatly different, while the feet are basically in a flat state, and the difference between the various modes is not large, so the static sensitive features are mainly the angles with the legs. Acceleration information is relevant. In the dynamic mode, the thigh, calf, and foot are all performing periodic movements, so the dynamic sensitive features involve three parts of the thigh, calf and foot movement information. The distance evaluation factors of each feature are sorted, and finally 20-dimensional static sensitive features and 40-dimensional dynamic sensitive features are selected according to the distance evaluation factors which are shown in Table A1 in Appendix A.

A three-layer BP neural network structure, with the neurons of the input layer, the hidden layer, and the output layer with labelled i, j, and k, respectively [27]. The parameter settings of each neural network in the recognition model are shown in Table 3. The function of the first neural network is to distinguish between dynamic and static data. As the recognition difficulty is low, so the number of input nodes was set to 5. The number of hidden layer nodes is 25. The input of the static neural network is a sensitive 20-dimensional feature, so it contains 20 input nodes, and the hidden layer was set to 100 nodes. The number of input layer nodes in the dynamic neural network was 40. Since there are many recognition classes and many similar gaits, it is difficult to identify. So, the number of hidden layers was set to 200.

### 3.3. Experimental and Applied Research

#### 3.3.1. Single Motion Pattern or Gesture Recognition Experiment

Multiple healthy subjects were selected to participate in the gait pattern recognition experiment. Subjects were required to wear a gait analysis system to perform data acquisition experiments in a single exercise mode. Each exercise mode was subjected to 10 sets of experiments. Among them, the metronome performs the step frequency control during fast walking, slow walking and constant speed walking to ensure the stability of the speed in the three sports modes. The running mode is performed on a treadmill with a uniform running setting function. The height of the step was fixed at 15 cm, the width was fixed at 29 cm, and the slope angle is 20°. Subjects walked for 1 minute before data acquisition to accommodate the metronome walking speed to reduce abnormal fluctuations in data during the acquisition process.

The collected data was separately pre-processed and windowed and intercepted, wherein the window length $T_{w}=64$ samples was taken, and the stacking time $T_{s}=32$ samples was taken. A sample set of machine learning training data was obtained, where in the independent variable is a feature extraction result of the data in the window, and the dependent variable is an experimental data tag value. Because the data was processed in groups, the data needed to be stratified and sampled, and 75% of the data was randomly selected as the training set. The experimental results are recorded in Table 4. It can be seen that the accuracy of the static neural network test group was 93.57%, the accuracy of the dynamic neural network test group was 100%, and the overall recognition rate reached 98.28%. In general, the single motion pattern recognition accuracy in the laboratory environmesnt was high, which verifies the reliability and effectiveness of the model.

Table 5 shows the results of the static neural network in the form of a confusion matrix. The data on the diagonal of the confusion matrix is the number of samples that are accurately identified, and the data outside the diagonal is the number of samples that are misidentified.

#### 3.3.2. Recognition Experiment of Mixed Motion Mode

A good recognition effect is obtained for a stable single motion mode. However, in the movement of the actual terrain, multiple modes are often switched, so the recognition experiment of the mixed motion mode is required. Multiple experimenters are required to sequentially perform multiple motion modes without any interruption in a certain order. Each dynamic motion mode is based on the terrain condition, and was manually timed by a stopwatch to control the walking speed of the subject according to the metronome.

Two sets of mixed motion experiments were designed based on the existing terrain. The first set of experiments was set up for horizontal pavement and stair mixing. The subjects transitioned from horizontal ground speed to single step upstairs, followed by continuous upstairs and two regular speeds. The process of the first set of mixed motion experiments is shown in Figure 11a.The second set of experiments was for horizontal pavement and slope. Mixing, the subject moved from standstill to the lower slope, then transitioned to the horizontal ground at a constant speed, stood still before the rising slope, and then climbed up the slope. The process of the second set of mixed motion experiments is shown in Figure 11b. In these experiments, each of the stairs was 29 cm wide and 15 cm high, and the slope inclination was 20°.

The collected data is preprocessed and windowed and intercepted, and the eigenvalues are calculated. The input is sent to the trained model in the single motion pattern recognition experiment, and the classification result of the mixed motion mode is obtained. The standard value is obtained from the result recorded by the stopwatch. The mode of the transition phase is determined by the proportion of the duration of the two modes falling within the window, and the mode with the longer time in the window is taken as the mode of the transition phase.

The experimental results of the mixed sample of horizontal pavement and stair pavement are shown in Figure 12. Figure 12a shows the variation curve of the plantar inclination angle of the left and right foot during the movement. The initial stage is the horizontal ground constant speed walking, and the left and right soles can be seen. The dip angle waveform is similar and the peak size is close, indicating that the left and right foot symmetry is very good. For the transition to a single step upstairs, on the one hand, due to the terrain climb, the peak of the plantee dip is significantly reduced, on the other hand, the left and right foot movements show significant differences. With the right leg as the active leg, the right foot sole inclination peak is significantly greater than the left foot sole inclination peak. During the continuous climbing up the stairs, the left and right legs alternately act as the forward active legs, and the waveform of the inclination of the left and right soles of the feet is restored to symmetry, and finally the two steps are taken at a constant speed.

Figure 12b is a comparison diagram of the recognition result and the standard value. Referring to Table 1, it can be seen that the constant speed walking label is 6, the single step upper step label is 12, and the continuous upper step label is 10. The number of windows is 39, and the number of misidentifications is one. The recognition accuracy is 97.4%.

The experimental results of the horizontal pavement and the slope pavement are shown in Figure 13. The curve of hip angle during exercise is shown in Figure 13a. The hip joint angle fluctuation during the stationary standing phase is small. When the thigh is forward and the hip joint angle is increased, began to enter the lower slope stage. The last 14 gait cycles are the lower slope process, and the lower slope phase is limited by the terrain. To ensure the stability of the body, the step size was kept small, so the hip joint flexion angle is also small. While entering the horizontal road at constant speed, the stride is obviously increased, and the hip joint peak increases in a single gait cycle. At the same time, the pitch frequency has also increased, and the duration of a single gait cycle has been shortened. While completely stopping in front of the rising slope, the hip joint peak at the last step of the horizontal road at constant speed is significantly smaller. The slope is started after standing still for six seconds in front of the slope. During the motion on the upward slope, the hip flexion peak increased to the ground uplift, and the entire experiment was completed. Figure 13b is a comparison of the recognition result and the standard value. Referring to Table 1, the stationary standing label is 1, the lower slope label is 15, the normal speed walking label is 6, and the upper slope label is 14. The number of windows is 97, the number of misidentifications is 7. The overall recognition accuracy is 92.7%, and the recognition accuracy is high in the stable phase.

## 4. Discussion

It can be seen that the difference between the static and dynamic modes is large, and the recognition model can distinguish between the two well. The categories of misidentification in the static mode mainly appeared between stationary standing and standing while carrying weight. There were a total of 114 groups of standing still samples, 88 of which were correctly identified and 26 were misidentified as weight standing. There were 124 groups for standing with weight, of which 123 were correctly identified. One group was misidentified as standing without load as there is no significant difference between the motor features such as the lower limb joint angle of the two postures, the difference is only reflected in the plantar pressure. The mass of the weight was lighter than the mass of the human body. After the mass was increased, the pressure of the plantar pressure was not much different. In addition, the interference of data fluctuations leads to a certain room for improvement in the identification of these two modes either by increasing the number of plantar pressure sensors or by replacing the pressure sensor with higher sensitivity. Regarding the single pattern recognition, the accuracy rate of static neural network recognition was lower than that of dynamic recognition network, because of large difference between dynamic patterns, and small difference between static patterns. For example, it is difficult to distinguish between standing still and bearing weight. In the dynamic mode, the test group data in the stable single mode showed high recognition accuracy. Due to the complementarity of various features, the behaviors of fast and slow walking, walking upstairs and running can be realized with accurate identification.

In the mixed motion experiments on horizontal sidewalks and stairs, a sample point of constant speed walking was identified as slow walking. In the mixed motion experiments on slope and horizontal road, a stationary standing sample was misidentified as a weighted standing, a downhill sample was misidentified as slow walking, a constant speed walking sample was misidentified as downhill, one sample walking at constant speed was misidentified as slow walking, two stationary standing samples were misidentified as weighted standing, and one uphill sample was misidentified as a single step up the step.

According to the two sets of mixed motion mode experiments, it can be seen that the misidentification mainly occurred in the mode switching phase, and the misidentification in the static to dynamic switching phase is far more than the misidentification in the dynamic mode switching phase. There is a short period during transition from the standstill to the start of the movement which causes fluctuations in the pressure of the foot, causing the static standing to be misidentified as a weight standing before the static switch to dynamic mode. The sliding window in the static to dynamic switching phase had both a stable static phase and a dynamic phase with large fluctuations. The mean and variance extracted in the window were inevitably far from the preset mode, resulting in a far-reaching. The occurrence of misidentification points, such as the transition phase of standing still to the upper slope in the second set of mixed motion mode experiment, was recognized as a single step.

In this research area, using a single *three-axis* accelerometer to achieve a recognition rate of 99% for a single motion state in four motion modes: lying, standing, walking, and running [10]. The recognition rate of only 6 types of sports modes including sitting, standing, walking, obstacle crossing, and going up and down using the sole force sensor is 98.8% [11]. Using 64 photoelectric matrix insoles and exoskeleton sensors to achieve 7 types of sports modes: ground-level walking, stair ascending, stair descending, sitting, standing, sit-to-stand and stand-to-sit, recognition rate 99.4% [12]. The combination of EMG and prosthetic mechanical sensors has achieved recognition rates of 97.7% and 97.8%, respectively, in the three scenarios of flat walking, stairs, and ramps [15,16].

In this study, we use six gyroscopes and six pressure sensors to collect the data of human body’s posture and motion, which provides convenience for integrating this system into exoskeleton system in the future. EMG acquisition system is not stable, skin temperature changes and sweating will affect the acquisition data, which is not conducive to the application of integration with exoskeleton in daily scenes. We propose 15 common human body postures and motion modes in exoskeleton application scenarios. The recognition accuracy of single motion mode is 98.28%, and that of mixed motion mode is 92.7% and 97.4%, respectively. Although the recognition rate of our system is not the highest at present, which is related to the 15 kinds of human body states and motion modes proposed by us. Compared with the simpler motion scenes mentioned above, the data volume of our multiple scenes increases the difficulty of data training and recognition. After we improve the experimental system and algorithm in the future, the recognition rate will be improved. At the same time, compared with the use of photoelectric matrix pressure insole [12], we only select the most important three positions of the foot to arrange the pressure sensor, and the cheap scheme is more feasible in practical application. In fact, the foot segmentation in our system, which can adapt to different terrain, is an initial design for exoskeleton. The gyroscope can be assembled into the leg of exoskeleton to realize system integration. Of course, our system also has shortcomings, the human foot and the exoskeleton foot can not fit closely at all times, which may cause false recognition in the complex road. The gyroscope sensor is also prone to dislocation between the legs, which affects the data acquisition. In the future, we will make up for the data of various sensors to reduce the generation of false identification. In addition, the entire wearable experimental system had too many wiring harnesses, which may interfere with the subject’s movement during the experiment and affect the accuracy of the experiment. If misidentification occurs during the use of the exoskeleton, it will affect the exoskeleton robot to switch the correct control mode, causing the exoskeleton to run smoothly and even cause an wearer injury. And it is necessary to further improve the motion pattern recognition performance of the system. On the one hand, it is necessary to increase the number of samples to improve the generalization ability. On the other hand, it is necessary to increase the dynamic to static switching mode.

## 5. Conclusions

In order to improve the accuracy of the movement state recognition of the lower extremity exoskeleton robot, this paper proposes a motion state recognition model based on feature evaluation and a multi-layer BP neural network. Firstly, 15 common gait patterns and postures were determined from the actual use requirements of an exoskeleton. The multi-source lower limb motion information extraction; multi-dimensional time domain features, frequency domain features, and energy features were combined to form a joint state for different motion modes. The evaluation factors of each feature selected the sensitive features of each layer as the input of the BP neural network according to the size of the evaluation factor, thus establishing a nonlinear mapping model between the feature quantity and the motion state. The final experimental results showed that the model proposed in this paper can identify the wearer’s motion state with high precision, and it also obtains satisfactory results for the continuous motion recognition problem of multiple motion modes. The recognition accuracy rate in single motion mode is up to 98.28%, and the recognition accuracy of two groups in mixed motion mode is 92.7% and 97.4%, respectively.

In future work, we will complement the data collected by this wearable system with data collected by other intent recognition systems (such as EEG or EMG) to improve the recognition of more postures and movement states of the body, especially continuous and transitional movements. And the results of this research will be used in the multi-mode control of lower extremity exoskeleton to further explore the optimization of human machine coordinated motion in clinical.

## Figures and Tables

**Figure 1 sensors-20-00537-f001:**
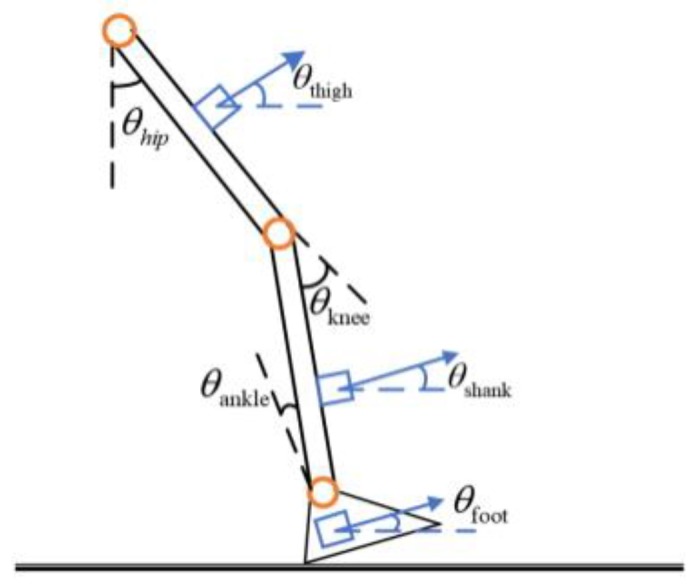
Leg inertial sensor arrangement.

**Figure 2 sensors-20-00537-f002:**
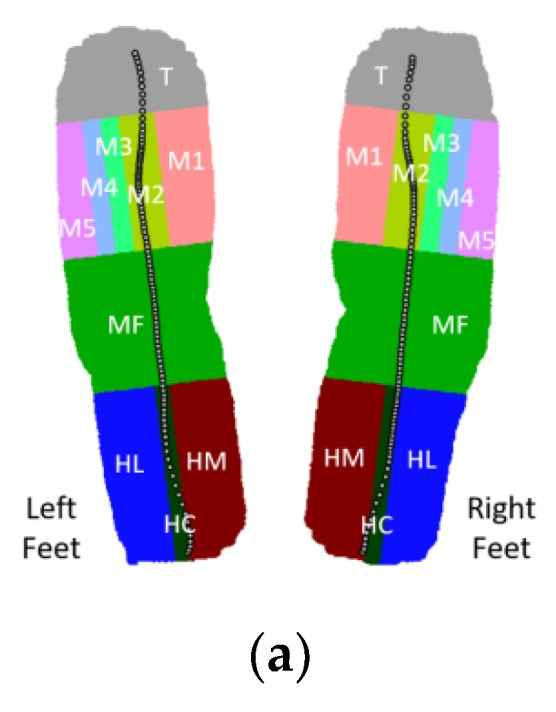
(**a**) Plantar pressure distribution; (**b**) Pressure curve of various regions of the sole.

**Figure 3 sensors-20-00537-f003:**
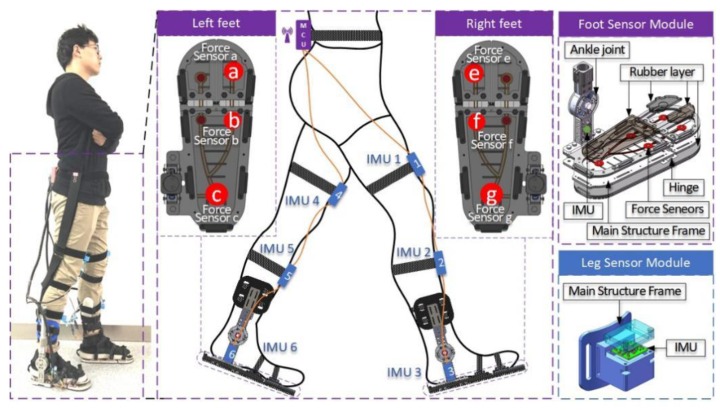
Design of wearable gait analysis system.

**Figure 4 sensors-20-00537-f004:**
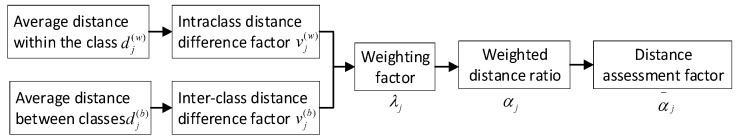
Feature screening process.

**Figure 5 sensors-20-00537-f005:**
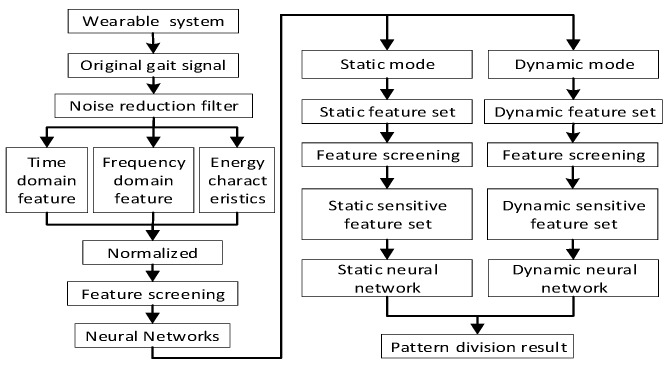
Motion pattern recognition model training process.

**Figure 6 sensors-20-00537-f006:**
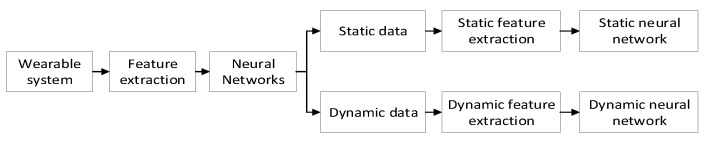
Motion pattern recognition model test flow.

**Figure 7 sensors-20-00537-f007:**
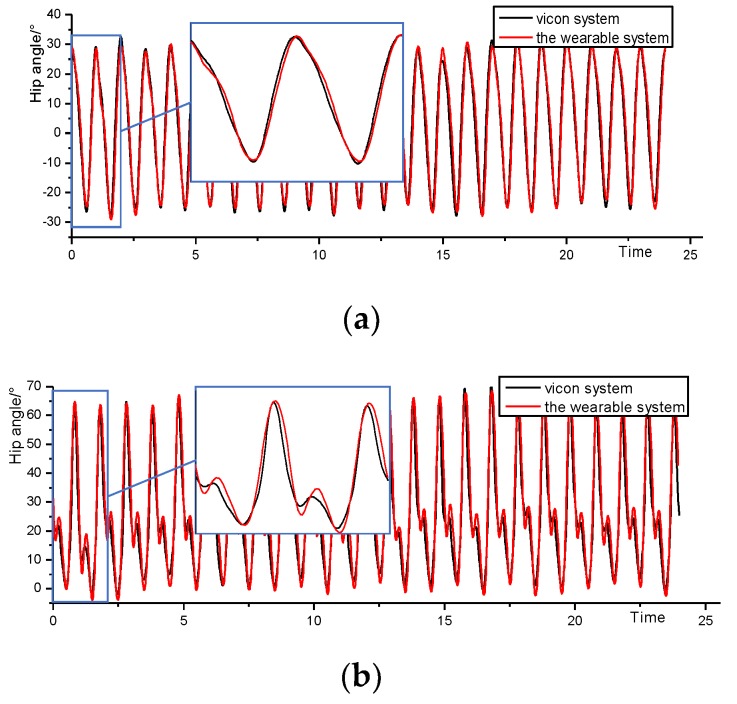
(**a**) Hip angle comparison curve; (**b**) Knee angle comparison curve.

**Figure 8 sensors-20-00537-f008:**
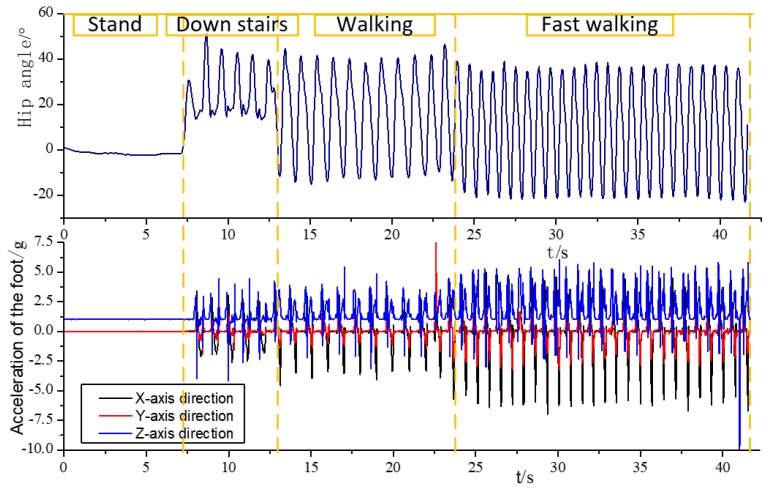
Hip joint angle and changes in the *three-axis* acceleration of the typical behavior.

**Figure 9 sensors-20-00537-f009:**
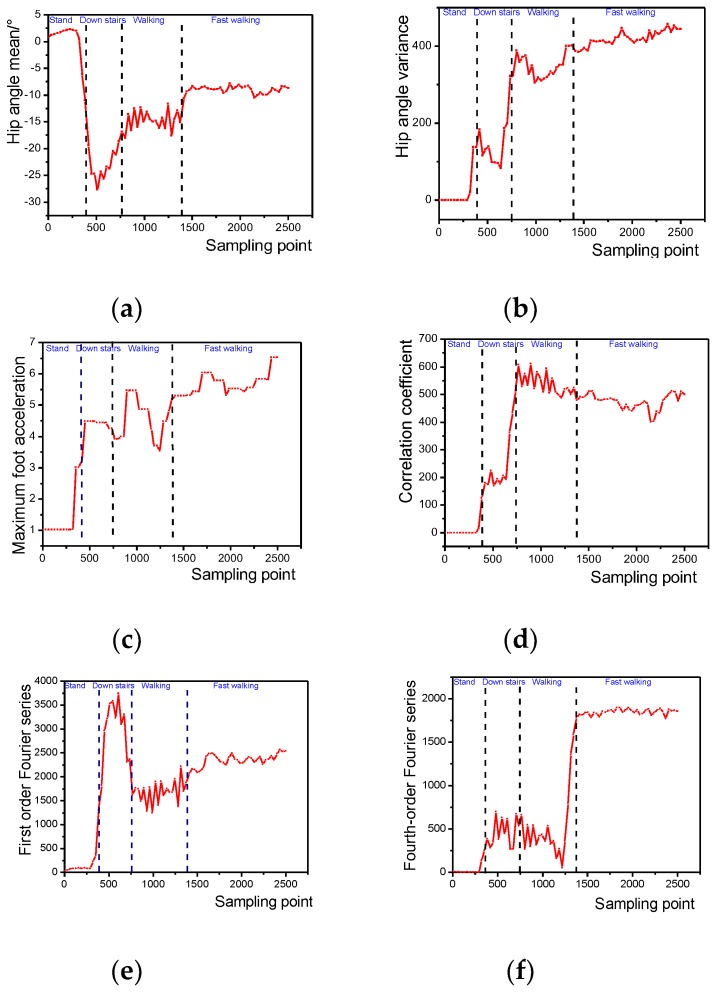
Comparison of the selected features of typical behavioral actions such as standing, stepping down, constant speed walking and fast walking. (**a**) Hip joint mean; (**b**) Hip joint angle variance; (**c**) Foot *x*-axis acceleration maximum; (**d**) Right knee angle correlation coefficient; (**e**) Knee joint angle first-order Fourier series; (**f**) Knee joint angle fourth-order Fourier series; (**g**) Plantar acceleration signal amplitude; (**h**) Hip joint angle wavelet entropy.

**Figure 10 sensors-20-00537-f010:**
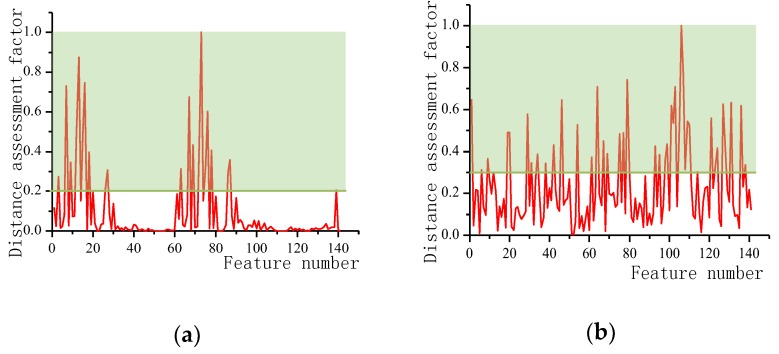
(**a**) Distance assessment factor for static feature sets; (**b**) Distance assessment factor for dynamic feature sets.

**Figure 11 sensors-20-00537-f011:**
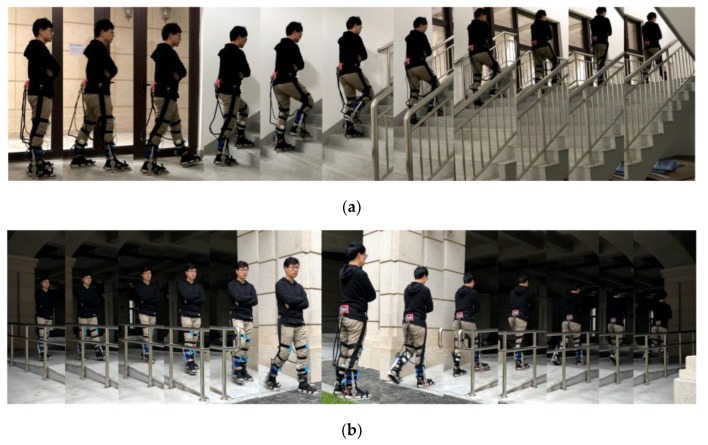
Two sets of experimental scenarios, (**a**) Mixed motion on horizontal sidewalks and stairs; (**b**) Slope and horizontal pavement mixing.

**Figure 12 sensors-20-00537-f012:**
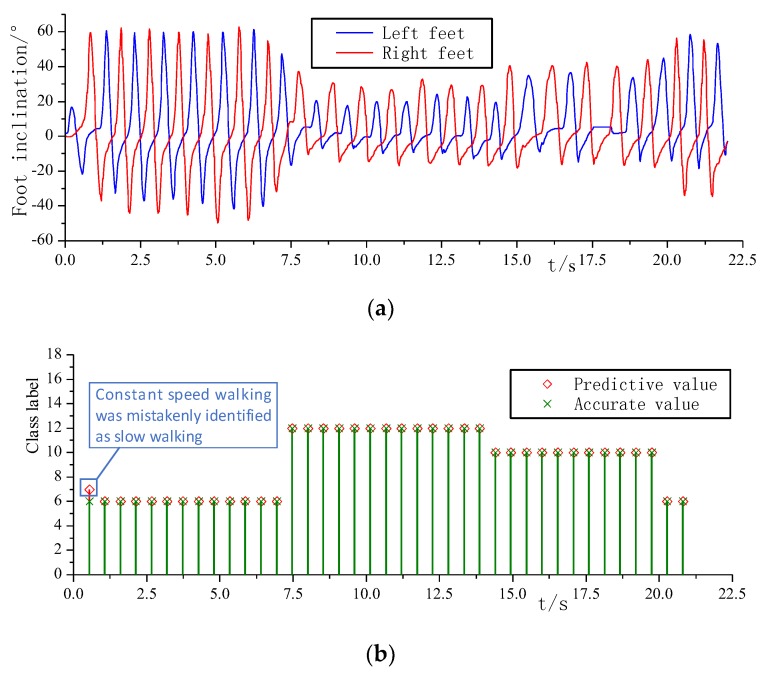
(**a**) Foot inclination during horizontal road and stair movement; (**b**) Identification results during horizontal road and stair movement.

**Figure 13 sensors-20-00537-f013:**
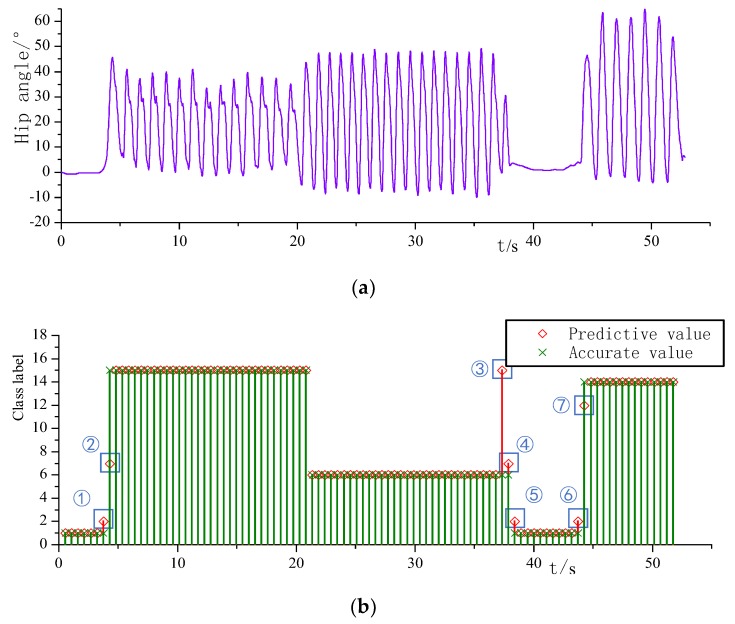
(**a**) Hip angle of mixed motion on flat and sloped roads; (**b**) Identification results of mixed motion on flat and sloped roads.

**Table 1 sensors-20-00537-t001:** Selected gait patterns.

Category	Condition	Gait Patterns	Detailed Descriptions	State Graph	Label
Static posture	Flat road	Standing still	Standing vertically	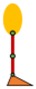	1
Standing with weight	Load 5 KG, stand still	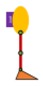	2
Sitting	Sit down, two calves vertical ground	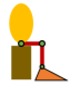	3
One knee down	Left leg bent, right knee touchdown	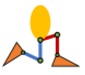	4
Dynamic attitude	Flat road	Fast walking	Walking speed is 4.5 km/h	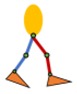	5
Constant speed walking	Walking speed is 3.0 km/h	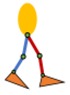	6
Slow walking	Walking speed is 2.0 km/h	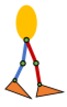	7
Walking in place	Step frequency is 1.0 hz	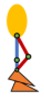	8
Jogging	Running speed 6.0 km/h	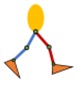	9
Stepped pavement	Continuously stepping up	Two legs alternately as front legs	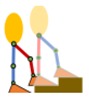	10
Continuously stepping down	Two legs alternately as front legs	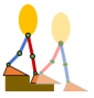	11
Single step step up	Right leg as a forward leg, left leg follows	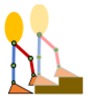	12
Single step step down	Right leg as a forward leg, left leg follows	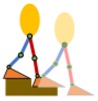	13
Slope road	Uphill	Constant slope, constant speed walking	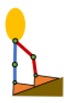	14
Downhill	Constant slope, constant speed walking	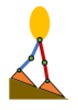	15

**Table 2 sensors-20-00537-t002:** 141-dimensional feature name and serial number.

Number	Feature	Feature Content
1–30	Mean	Average of the angle of the hip, knee, and ankle; (6 dimensions) Average of the three-axis acceleration of the hip, knee, and ankle; (18 dimensions) Average of pressure on the left and right feet. (6 dimensions)
31–60	Variance	The variance of the hip, knee and ankle angles of the two legs; (6 dimensions) The variance of the triaxial acceleration of hips, knees and ankles of two legs; (18 dimensions) Variance of three pressure points on both feet. (6 dimensions)
61–120	Maximum value	Maximum and range of angles for hips, knees, and ankles; (12 dimensions) Maximum and range values of triaxial acceleration of hip, knee and ankle; (36 dimensions) Maximum and range values of the three pressure points of left and right feet. (12 dimensions)
121–128	Correlation coefficient	Hip and knee angle correlation coefficient of the left leg and right leg; (2 dimensions) Knee and ankle angle correlation coefficient of the left leg and right leg; (2 dimensions) Acceleration coefficient of left foot and right foot in x-z plane; (1 dimension) Angle correlation coefficient between left hip and right hip; (1 dimension) Angle correlation coefficient between left knee and right knee; (1 dimension) Angle correlation coefficient between left ankle and right ankle joint. (1 dimension)
129–138	Fourier series	Fifth-order Fourier series of hip and knee angles of left leg. (10 dimensions)
139–140	SMA	The amplitude of the left and right foot acceleration signals. (2 dimensions)
141	Wavelet energy entropy	Wave energy entropy of left hip joint angle. (1 dimension)

**Table 3 sensors-20-00537-t003:** Neural network parameter settings.

	First Network	Static Neural Network	Dynamic Neural Network
Input layer	5	20	40
Hidden layer	25	100	200
Output layer	1	1	1

**Table 4 sensors-20-00537-t004:** Identification accuracy of each layer of neural network.

Identification Project	First Neural Network	Static Neural Network	Dynamic Neural Network	Overall Model
Training	100%	100%	100%	100%
Test	100%	93.57%	100%	98.28%

**Table 5 sensors-20-00537-t005:** Static neural network confusion matrix.

	1	2	3	4	5	6	7	8	9	10	11	12	13	14	15
**1**	88	1													
**2**	26	123													
**3**			84												
**4**				98											
**5**					104										
**6**						84									
**7**							125								
**8**								112							
**9**									104						
**10**										97					
**11**											87				
**12**												127			
**13**													126		
**14**														95	
**15**															91

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
