# Peer review of "Human Body Mixed Motion Pattern Recognition Method Based on Multi-Source Feature Parameter Fusion"

_sensors, 2020, doi:10.3390/s20020537_

Round 1

Reviewer 1 Report

The paper entitled “Human body mixed motion pattern recognition method 2 based on multi-source feature parameter fusion” presents an approach to perform an online classification of 15 gait and posture states based on the measurements provided by lower limb IMU and distributed foot sole sensors.

General comment:

The paper suffers for a weak document structure (in particular repeated mixes between “method” elements and “results” content) which make the reading and understanding very difficult.

The paper should be structured according to standard sections:

Introduction (Theory if needed) Experimental methods Subjects Procedure Data analysis -> feature extractions Results Discussion / Conclusion There is no description of the sensorised foot sole. Acco0rding to figure 3 it is difficult to know if, for example, the sole holding the sensor is rigid, or soft. Please provide details about the embodiment. There are many sentences that are not grammatically correct.

Specific comments:

For specific comments please refer to the attached and annotated pdf.

Author Response

First of all, thank you very much for your valuable comments on my article, which has helped me a lot to modify it.

Regarding the structure of the article, I agree with you so much, so I made changes one by one according to the comments you gave me in the PDF file, and I highlighted the revised content.

About the sole in the experimental equipment, it is made of metal frame and rubber material. In the revised version, I have described it in detail with text and pictures.

Attached is the revised file. 

Thank you!

Reviewer 2 Report

This paper presents the motion state recognition model using the multi-layer BE neural network. The authors defined the 141-dimensional parameter for the pattern recognition and extracted the model parameters for several unit patterns. The resulting recognition accuracy is very high over 90 %, so this paper provides valuable results to Sensors society. However, several concerns should be resolved before publication.

In the introduction part, the authors claim the paper originality in that a total 15 of categories are introduced regarding the gait pattern to evaluate the pattern recognition method. In the experiment shown in Figure 11, only several categories including the label 1, 6, 10, 12, 14, and 15 were tested to get the identification results. To improve the originality, the test results should be presented for more categories. Several previous researches are mentioned in the introduction part. The recognition results shown in Figures 12(b) and 13(b) need to be compared with the previous researches. In Table 4, the accuracy is 100 % for the dynamic neural network while 93.57 % for the static neural network. It is reasonable to assume that the accuracy is higher for the static neural network because static situation is much easier to recognize than the dynamic one. The reason of the higher accuracy for the dynamic neural network should be explained. Vicon pressure treadmill was used to measure the pressure distribution over the foot. A reference needs to be used to provide the information about the treadmill.

Author Response

Firstly, thank you so much for your comments, which have greatly helped my article revision.

Regarding the 15 gait patterns and postures mentioned in this article, we first identified each pattern. After that, we performed two sets of mixed recognition experiments on patterns 6, 10, 12 and patterns 1, 6, 14, 15. Therefore, a total of 15 models are involved.

Regarding single pattern recognition, the accuracy rate of static neural network recognition is lower than that of dynamic recognition network, because the difference between dynamic patterns is large, and the difference between static patterns is small. For example, it is difficult to distinguish between standing still and bearing weight. 

The specific model of the Vicon device has been added in the modified version.

Attached is a revised version of this article.

Thank you.

Reviewer 3 Report

The manuscript entitled „ Human body mixed motion pattern recognition method based on multi-source feature parameter fusion” is presented in an intelligible fashion. This paper studies the human body mixed motion pattern recognition technology based on multi-source feature parameters. The authors concluded that the recognition accuracy in single motion mode can reach up to 98.28%, while the recognition accuracy of the two groups of experiments in mixed motion mode was found to be 92.7% and 97.4%, respectively. The feasibility and effectiveness of the model were verified. I believe that the topic taken by the authors is interesting and current. I highlight positive point of the paper: based on the actual requirements of exoskeleton per use, 15 common gait patterns were determined. In despite of that, the paper cannot be published unless a minor revision is done, addressing the following points:

Introduction section: The text in in the following paragraph is a bit confusing, it should be revised and rewritten "The recognition result can be used as a control signal when the robot motion mode is switched, which increases the flexibility of the handover, and can also be used as a correction for the intentional perception result". Methods section: Information about the validity of your research tools is missing. Discussion section. Authors did not discuss with many more recent papers on this subject area. I agree with the authors that it is necessary to further improve the motion pattern recognition performance of the system: 1) it is necessary to increase the number of samples to improve the generalization ability, 2) it is necessary to increase the dynamic to static switching mode. I was pleased to see that they have highlighted this matter in the discussion. I suggest that some information about the strengths and weaknesses of this paper should be added. Conclusion section: I suggest that the authors add a paragraph stating how their findings can be incorporated in the everyday clinical practice. Overall the paper is good and has potential to contribute to the field of knowledge.

Author Response

Thank you very much for your encouragement and valuable comments, I have modified the article according to each of your comments. For details, please see the revised version. (I highlighted the revised content.)

Round 2

Reviewer 2 Report

In the review comments, I asked the authors to present accuracy comparison between the proposed method and the previously proposed ones. This comparison is important to show originality of the proposed method. However, the revised manuscript does not present the comparison results without any explanation. 

Author Response

Thank you very much for your encouragement and valuable comments. I have modified the article according to your comments and added my comparison with other methods in this field. For details, please see the revised version. (I highlighted the revised content.)
